# Thioredoxin/Glutaredoxin Systems and Gut Microbiota in NAFLD: Interplay, Mechanism, and Therapeutical Potential

**DOI:** 10.3390/antiox12091680

**Published:** 2023-08-28

**Authors:** Minghui Zhu, Omer M. A. Dagah, Billton Bryson Silaa, Jun Lu

**Affiliations:** Engineering Research Center of Coptis Development and Utilization/Key Laboratory of Luminescence Analysis and Molecular Sensing, Ministry of Education (Southwest University), College of Pharmaceutical Sciences, Southwest University, Chongqing 400715, China; 1843476a@gmail.com (M.Z.); omer20062010@gmail.com (O.M.A.D.); brysonbilton@gmail.com (B.B.S.)

**Keywords:** thioredoxin, reactive oxygen species, NAFLD, gut microbiota dysbiosis, oxidative stress

## Abstract

Non-alcoholic fatty liver disease (NAFLD) is a common clinical disease, and its pathogenesis is closely linked to oxidative stress and gut microbiota dysbiosis. Recently accumulating evidence indicates that the thioredoxin and glutaredoxin systems, the two thiol-redox dependent antioxidant systems, are the key players in the NAFLD’s development and progression. However, the effects of gut microbiota dysbiosis on the liver thiol-redox systems are not well clarified. This review explores the role and mechanisms of oxidative stress induced by bacteria in NAFLD while emphasizing the crucial interplay between gut microbiota dysbiosis and Trx mediated-redox regulation. The paper explores how dysbiosis affects the production of specific gut microbiota metabolites, such as trimethylamine N-oxide (TMAO), lipopolysaccharides (LPS), short-chain fatty acids (SCFAs), amino acids, bile acid, and alcohol. These metabolites, in turn, significantly impact liver inflammation, lipid metabolism, insulin resistance, and cellular damage through thiol-dependent redox signaling. It suggests that comprehensive approaches targeting both gut microbiota dysbiosis and the thiol-redox antioxidant system are essential for effectively preventing and treating NAFLD. Overall, comprehending the intricate relationship between gut microbiota dysbiosis and thiol-redox systems in NAFLD holds significant promise in enhancing patient outcomes and fostering the development of innovative therapeutic interventions.

## 1. Introduction

Non-alcoholic fatty liver disease (NAFLD) is a clinical–pathological syndrome characterized by excessive deposition of fat and fatty degeneration in liver cells in the absence of excessive alcohol consumption or other known causes of liver disease. Pathologically, NAFLD is generally categorized into two types: simple fatty liver and non-alcoholic steatohepatitis (NASH) and can progress to fatty liver fibrosis and even liver cancer. Besides the direct harm from liver disease, NAFLD patients are at significantly increased risk of developing cardiovascular diseases, type 2 diabetes, and chronic renal insufficiency [1,2]. However, the specific pathogenesis of NAFLD/NASH remains unclear, and there is currently no specific treatment available. Clinical pharmacological research on NAFLD is actively underway.

NAFLD is a disease closely related to various factors, including genetics, environment, and diet. To emphasize its metabolic nature, in March 2020, an international expert group proposed renaming it as Metabolic Associated Fatty Liver Disease (MAFLD) [3]. When NAFLD progresses to NASH, the risk of liver fibrosis and liver cancer increases significantly. Consequently, in the natural development of NAFLD, the progression of the disease can be effectively stopped by improving NASH. The exact cause of NAFLD/NASH is still unknown, and the “two-hit” hypothesis proposes that insulin resistance in peripheral adipose tissue results in the breakdown of fats, elevated levels of free fatty acids in the bloodstream, and excessive transportation of fatty acids to the liver, ultimately leading to hepatic steatosis [4,5]. This is the “first hit”. Prolonged and excessive fat accumulation in the liver cannot be eliminated, causing mitochondrial dysfunction. This dysfunction leads to endoplasmic reticulum stress, oxidative stress, and the release of inflammatory factors, which exacerbate hepatocellular damage and contribute to the development of NASH from simple fatty liver disease, referred to as the “second hit”. Reduced secretion of adiponectin (ADPN) and elevated pro-inflammatory factors in peripheral adipose tissue further promote inflammatory responses and worsen insulin resistance, creating a vicious cycle [4]. Additionally, liver cell apoptosis is critical for the development of NASH. The pathogenesis of NAFLD/NASH is highly complex, and the current understanding has shifted from the “two-hit” to the “multiple parallel hits” theory, involving diverse factors such as oxidative stress, gut microbiota, insulin resistance, adipokines, genetics, and epigenetics, and others [5,6]. This review focuses on the role of thiol-redox imbalance and gut microbiota dysbiosis in the pathogenesis and the therapeutic potential of NAFLD. Particularly, the thioredoxin/glutaredoxin (Trx/Grx) systems, which are the key cellular disulfide reductase system, play critical roles in NAFLD and may act as molecular targets for NAFLD, providing insight for the treatment of NAFLD.

## 2. Disruption of Thiol Homeostasis in NAFLD

### 2.1. Sites of ROS Production

Reactive oxygen species (ROS) are inevitable byproducts of aerobic metabolism in living organisms, including superoxide (O_2_^−^), hydrogen peroxide (H_2_O_2_), hydroxyl radicals (HO·), and so on. Among them, hydroxyl radicals are the most reactive, capable of attacking almost all biological molecules, inducing lipid peroxidation, and causing DNA double-strand breaks, making them the most destructive type of ROS [7,8], Superoxide (O_2_^−^) serves as the precursor of other ROS. When O_2_ leaks from the mitochondrial respiratory chain, an O_2_ molecule accepting an electron generates O_2_^−^, which can be dismutated by superoxide dismutase (SOD) into H_2_O_2_ and O_2_. H_2_O_2_ is not a free radical molecule but can be catalyzed to generate highly active HO· through the Fenton reaction when accepting an electron. The majority of ROS in cells originate from this process [9,10]. The excess ROS causes damage to mitochondrial DNA (mtDNA), respiratory chain complexes, lipids, and other components [11], which, in turn, further induces ROS production, creating a vicious cycle [12,13].

The other source sites of ROS include the endoplasmic reticulum (ER), a crucial site for protein synthesis, post-translational modification, processing, and folding in cells. In the ER lumen, the correct folding of most proteins requires the formation of disulfide bonds between cysteine residues to stabilize their structures. The formation of disulfide bonds is a reversible process that can be achieved through thiol–disulfide interchange. In eukaryotic cells, the folding of oxidative proteins is mainly catalyzed by a series of redox enzymes, such as protein disulfide isomerase (PDI), a member of the thioredoxin superfamily proteins [14]. Additionally, the folding of ER proteins is also dynamically regulated by the balance of the redox buffer, such as GSH/GSSG, which maintains the cellular redox state in a reduced state, GSH accounts for approximately 99% of the overall content, whereas it is found in only 50–60% in the endoplasmic reticulum [15]. PDI transfers two electrons to its substrate ERO1 (ER oxidoreductase 1), producing hydrogen peroxide. This electron transfer mode indicates the induction of ROS production [16].

Moreover, peroxisome, the organelles present in all eukaryotic cells, is believed to be the site to produce ROS. Peroxisome is closely related to the regulation of liver lipid homeostasis [17]. Medium- and long-chain fatty acids (FAs) are oxidized predominantly in mitochondria, whereas very long-chain FAs (VLCFAs) are metabolized almost exclusively by β-oxidation in the peroxisome and are involved in α-oxidation of fatty acids [18]. Peroxisomes produce acetyl coenzyme A through fatty acid β-oxidation, which plays a role in lipid signal transduction to promote lipid autophagy [19]. It is estimated that peroxisomes contribute about 35% of ROS in cells.

Moreover, NADPH oxidase, also known as NOX, is the only enzyme whose main function is to produce ROS. Researchers have identified seven isoforms, NOX1, NOX2, NOX3, NOX4, NOX5, DUOX1, and DUOX2, of which NOX2 is highly expressed in the liver [20]. NOX2 consists of 6 subunits: gp91phox, p22phox, p47phox, p67phox, p40phox, and Rac. Activation of NOX2 requires the translocation of p47phox, p67phox, p40phox, and Rac from the cytoplasm to the cell membrane, where they bind to gp91phox and p22phox to form the complex to produce the oxidants [21]. NOX2 affects signal transduction and immune function by transferring electrons from NADPH to molecular oxygen and generating large amounts of oxidants, a process involved in a variety of physiological activities such as cell proliferation and differentiation [22]. Studies have reported that in liver-resident Kupffer cells and infiltrating macrophages, NOX2-derived ROS stimulate them to produce pro-inflammatory cytokines such as TNF-α, IL-6, and IL-1β in response to a variety of factors, including oxidized low-density lipoproteins (LDL) and lipopolysaccharides (LPS) [23,24,25]. Therefore, over-activation of NOX2 leads to a surge of oxidants, which directly activates downstream inflammatory pathways and indirectly damages DNA and regulates protein phosphorylation, causing oxidative damage and exacerbating inflammation, which leads to cellular dysfunction and the onset and deterioration of diseases and affects the function and metabolism of tissues and organs [26,27]. Moreover, increased NOX2 activity has been shown in patients with NAFLD [28,29], suggesting a correlation between NOX2 and NAFLD. As mentioned above, oxidative stress serves as the “second hit” in the pathogenesis of NAFLD and is typically accompanied by a large amount of ROS. ROS can lead to the inactivation of mitochondrial respiratory chain enzymes and GAPDH, inhibit Ca^2+^ channels on the membrane, and induce liver cell damage [30]. Furthermore, ROS can induce the production of cytokines and Fas, along with their ligands, exacerbating lipid peroxidation, thus aggravating liver inflammation and hepatocyte fibrosis [31]. Cellular FFA loading can lead to changes in mitochondrial adaptability, enhancing fatty acid oxidation and upregulating electron transport, further resulting in excessive ROS production [32]. Thus, the excessive production of ROS can damage mitochondrial proteins, lipids, and DNA, a process that may play a role in the initiation of NASH. In the late stage of NASH, mitochondrial ultrastructure damage affects its function, reduces ATP synthesis, and intensifies ROS overproduction [33].

### 2.2. Thioredoxin (Trx) System

When the liver is exposed to high levels of ROS or electrophilic reagents, oxidative damage is prone to occur. At this point, the body initiates a series of antioxidant defense mechanisms [34]. The mammalian thiol-dependent antioxidant system comprises the thioredoxin (Trx) system and the GSH-glutaredoxin (Grx) system, both of which undergo interconversion between thiol (-SH) and disulfide bond (-S-S-) forms to eliminate ROS and maintain the redox homeostasis in the body [35].

The Trx system mainly consists of Trx, thioredoxin reductase (TrxR), and nicotinamide adenine dinucleotide phosphate (NADPH). It is a major thiol-dependent antioxidant system in the body, responsible for maintaining intracellular redox homeostasis and protecting cells from oxidative stress (Figure 1) [9,36]. Mammals have two isoforms of Trxs. Trx1 is present in the cytoplasm with five cysteines, two active site cysteines, and three structural cysteines, while Trx2 is located in the mitochondria with only two cysteine residues in its active center [36]. The Trx system transfers electrons to various key cellular proteins, including peroxiredoxin (Prx), which directly scavenges H_2_O_2_, and methionine-S-sulfoxide reductase (MsrA), which participates in oxidative stress defense [35,37,38]. The proper functioning of the Trx system relies on TrxR accepting electrons from NADPH and transferring them to Trx. The activity and expression levels of TrxR directly determine the normal biological function of Trx and its downstream proteins. Mammalian cells have two forms of TrxR, cytoplasmic TrxR1 and mitochondrial TrxR2, both of which have FAD and NADPH binding domains as well as interface domains. At their N-terminus, they have an active site CVNVGC, and at their C-terminus, they possess a special sequence Gly-Cys-Sec-Gly (GCUG), with Sec being the catalytic active site of TrxR, which is necessary for its reduction activity [39].

Studies have shown significant changes in the Trx system in metabolic syndrome [50,51]. For example, in metabolic syndrome patients, consuming a diet rich in high-saturated fatty acids (HSFA) significantly increased the mRNA levels of Trx in adipose tissue. Simultaneously, postprandial adipose tissue TrxR1 mRNA significantly decreased.This suggests that the ability of Trx to revert from its oxidized form to its reduced form decreased, leading to a compensatory increase in Trx gene levels, indicating increased oxidative stress due to saturated fat intake [52]. The Trx system also plays a crucial role in adipocyte dysfunction and obesity [53]. In obese individuals with metabolic disorders, TrxR activity and Trx content in subcutaneous tissue are significantly increased because subcutaneous fat seems to provide good protection against increased oxidation in obese subjects compared to visceral fat. Protein levels of Trx, GPx, and CuZnSOD, as well as activities of GPx, GR, GST, TrxR, and SOD, were significantly higher in “at-risk” obese women than in metabolically healthy women [54]. While the expression of Trx-dependent peroxiredoxin (Prx3) in adipose tissue is significantly reduced [55], Prx3 is a key molecule regulating adipocyte oxidative stress and adipokine expression [56]. This may indicate that expression of various antioxidant enzymes is predominantly and specifically regulated by different transcription factors, but the detailed mechanism behind it needs further clarification.

Trx system in NASH has been shown to be changed in several studies [57,58]. Serum Trx levels are significantly increased in NASH patients compared to simple steatosis patients, making it a potential biomarker to distinguish NASH from early-stage fatty liver [59]. In animal models of choline-deficient liver steatosis, the activities of hepatic Trx1 and TrxR are upregulated at day 14 but significantly decreased at day 30 compared to day 14 [60]. Furthermore, the level of thioredoxin-interacting protein (TXNIP) in the liver of NAFLD patients is significantly elevated [61], and in mice fed a methionine and choline-deficient diet to induce NASH, the *TXNIP* gene was overexpressed, and expression of hepatic TrxR1 and TrxR2 decreased [62].

### 2.3. Glutathione (GSH)-Grx System

In addition to the Trx system, the glutathione-glutaredoxin (GSH-Grx) system is another crucial thiol-dependent redox system involved in cellular redox balance regulation [63]. It comprises NADPH, glutathione reductase (GR), and GSH coupled with Grxs, including cytosolic Grx1 and mitochondrial Grx2. The GSH-Grx systems mainly mediate redox signal transduction by reversibly catalyzing protein *S*-glutathionylation. GSH is the most abundant thiol small molecule, playing a vital role in maintaining intracellular redox homeostasis [64]. GSH can reduce various proteins, including glutathione peroxidases and glutathione transferases [65], and the resulting GSSG can be recycled by GR using NADPH as an electron donor [66].

Studies have shown that when *Glrx1* KO mice are given a standard diet for some time, they develop obesity, hyperlipidemia, and fatty liver, resembling NAFLD. When fed a high-fat diet, *Glrx1* KO mice develop fatty liver inflammation and progress to NASH [67]. In NAFLD animal models fed with a high-fat diet, *S*-glutathionylated protein levels increase, and an important target protein, Sirtuin-1 (SirT1), is identified. SirT1 is a NAD+-dependent class III histone deacetylase that regulates key transcription factors coordinating hepatic lipid metabolism [68,69,70]. Activation of SirT1 ameliorates Non-Alcoholic Fatty Liver (NAFL); conversely, hepatic SirT1 deficiency leads to steatosis [71], and thiol modification of SirT1 regulates lipid metabolism through acetylation of key transcription factors. S-glutathionylation inactivates SirT1 and promotes hyperacetylation and activation of downstream target proteins such as p53 and sterol regulatory element binding protein (SREBP) [68]. Activation of p53 is a cellular response to stress, resulting in the alteration of metabolism and the arrest of the cell cycle, and severe oxidative damage to hepatocytes may trigger p53 to induce cell death [72]. Studies also suggest that SirT1 overexpression can improve NAFLD [73], while Sirt1 knockout in mice leads to NAFLD (Figure 2) [74].

Grx2 has two known isoforms encoded by the same gene (*GLRX2*): Grx2a (mitochondrial) and Grx2c (cytoplasmic) [75,76]. The role and effects of mitochondrial Grx2-mediated glutathionylation in cellular metabolism are still poorly understood. A mouse model with a depletion of mitochondrial Grx2 fed with a standard diet had spontaneous weight gain and accumulation of lipid droplets in the liver, suggesting that Grx2 is involved in mitochondrial redox environment and lipid metabolism regulation [77]. Genes such as cytochrome P-450 7a1 (Cyp7a1), which is specifically expressed in the liver and participates in the reduction of cholesterol, were found to be significantly downregulated in the mice model [77].

### 2.4. Nrf2 Signaling Pathway

The Keap1-Nrf2-ARE pathway is one of the major pathways that maintain cellular homeostasis during oxidative stress in the liver and is also regulated by Trx. Its function is to protect cells from endogenous and exogenous damage caused by oxidative stress. When Nrf2 accumulates significantly in the cell nucleus, a large number of antioxidant and phase II detoxification enzyme genes are upregulated [78]. Previous studies have identified over 200 downstream target genes regulated by Nrf2 in human cells [79], including the genes encoding NAD(P)H quinone oxidoreductase 1 (NQO1), HO-1, GST, GPx, CAT, SOD, and GR (Figure 1) [80]; It controls the expression of the glutamate–cysteine ligase catalytic subunit (GCLC) and glutamate–cysteine ligase modifier subunit (GCLM) to maintain the proper ratio of GSH to GSSG in cells [81]; Nrf2 also positively regulates some enzymes involved in liver detoxification, including Trx1, TrxR1, Gpx2, and GST [82,83]. These detoxification enzymes can eliminate H_2_O_2_, free radicals, and oxidized thiols in the cytoplasm, endoplasmic reticulum, and mitochondria [84]. Thus, Nrf2 represents a potential therapeutic target for NAFLD.

In addition to Nrf2, which has been shown to bind to the ARE of the Trx, TrxR, and Prx1 promoters [85,86,87], hypoxia-inducible factor (HIF)-1α synergistically up-regulates specific genes during hypoxia with another class of transcription factors, the E26 translationally specific (Ets) family of related proteins [88]. In human PC3 prostate cancer cells, the Prx1 gene promoter was significantly induced by either H_2_O_2_ or reoxygenation after hypoxic growth [89]. Co-transfection of constructs overexpressing Ets-1 or Ets-2 with the Prx1 promoter construct resulted in increased promoter activity, and ChIP assays also confirmed that both Ets-1 and Ets-2 bound to the Prx1 gene promoter in PC3 cells [89]. Thus, it can be hypothesized that the Ets pathway may play a role in regulating redox control during hypoxia and reoxygenation to complement the activation of the Nrf2 pathway. In addition, specificity protein 1 (Sp1), Fas/Jun, TATA box binding protein (TBP), cAMP response element binding protein (CREB), retinoic acid receptor/retinoid X receptor (RAR/RXR), and other transcription factors bind to their DNA sites to regulate Trx [85,87,90,91]. Retinoid X receptor (RAR/RXR) and other transcription factors that bind to DNA sites to regulate Trx; Octamer binding protein (Oct-1), Sp1, Sp3, and other transcription factors bind to the DNA site to regulate TrxR [92]. However, whether these transcriptional regulations are involved in NAFLD/NASH is unclear.

## 3. Gut Microbiota in NAFLD

Over the past decade, extensive research has highlighted that imbalance in the gut microbiota, known as gut microbiota dysbiosis, plays a significant role in the development and progression of metabolic disorders like NAFLD, obesity, and T2DM [93,94,95]. The gut microbiota and its metabolites have been implicated in the pathogenesis of these conditions [96,97]. Accumulating evidence indicates that there is internal communication between the gut microbiota and the liver, forming the gut–liver axis. This communication is of significant importance in the occurrence and progression of liver diseases, including NAFLD [98]. The gastrointestinal tract houses trillions of microorganisms, which consist of a diverse community of bacteria, viruses, fungi, and others that interact with the host and the environment. Dysbiosis of the gut microbiota disrupts normal metabolic processes, leading to altered nutrient absorption, inflammation, impaired lipid metabolism, and insulin resistance, all of which are implicated in NAFLD [96,97].

### 3.1. Dysbiosis and Oxidative Stress in NAFLD

Dysbiosis in NAFLD can trigger oxidative responses, leading to inflammation and injury in the liver [99,100]. This highlights the interplay between gut microbiota dysbiosis and oxidative stress in the development of NAFLD, underscoring the importance of comprehensive approaches to address both factors for effective prevention and treatment. Oxidative stress interacts synergistically with other factors such as mitochondrial dysfunction, endoplasmic reticulum stress, altered lipid metabolism, and the generation of ROS as by-products during processes like fatty acid β-oxidation in peroxisomes, exacerbating liver injury in NAFLD [19,101]. These interactions result in elevated production of ROS and compromised antioxidant mechanisms, which involve various enzymes like catalase, SOD, and GPx, along with nonenzymatic compounds such as GSH and alpha-tocopherol [102]. In addition to oxidative stress, inflammation further exacerbates liver injury in NAFLD [100]. The redox-sensitive transcription factors Nrf2 and NF-κB are key regulators of the antioxidant system and inflammatory response, respectively. While Nrf2 activation mainly promotes antioxidant defense, NF-κB activation can have both protective and pro-oxidant effects (Figure 3) [100,103].

In patients with NASH, elevated levels of serum Trx have been observed compared to those with simple fatty liver, indicating its potential as a biomarker for distinguishing between the two conditions [104]. Higher Trx was found to be positively associated with the severity of NASH and increased iron accumulation in the liver. Trx levels, conversely, declined with the reduction of hepatic iron stores [59,105]. This could be attributed to ferroptosis, an iron- and thiol- associated oxidative stress process that impacts the pathogenesis of NASH [106,107,108]. Similarly, in animal models of NASH, alterations in Trx and TrxR expression were observed. NASH induction led to increased expression of the TXNIP gene and reduced expression of TrxR1 and TrxR2 in the liver. Oxidative stress triggers the upregulation of TXNIP expression by the activation of its promoter containing a carbohydrate response element (ChoRE), which is modulated by transcription factors including MondoA:Max-like protein X (MLx), nuclear factor Y (NF-Y), and carbohydrate response element-binding protein (ChREBP) [109,110].

### 3.2. Impact of Intestinal Barrier Dysfunction and Gut Microbiota Metabolites on NAFLD 

Numerous studies have consistently shown that individuals with NAFLD exhibit reduced microbial diversity in their gut. These studies, conducted in both mice and human subjects, have identified specific bacterial taxa that display differential abundance in NAFLD. Bacteria such as *Lactobacillus*, *Lachnospiraceae bacterium 609*, and *Barnesiella intestinihominis* were found to be associated with NAFLD, while others like *Allobaculum* and *Lactobacillus* exhibited altered levels [111]. Furthermore, there were changes in the overall gut microbiota composition, including an increase in Actinobacteria and a decrease in Bacteroidetes, among NAFLD patients compared to healthy controls [112]. Various bacterial taxa, including *Bradyrhizobium*, *Anaerococcus*, *Peptoniphilus*, *Propionibacterium acnes*, *Dorea, Ruminococcus*, and *Blautia*, showed varying levels of abundance in both NAFLD and non-alcoholic steatohepatitis (NASH) patients [112,113]. The gut microbiota affects NAFLD mainly through two main mechanisms: the disruption of the intestinal barrier and the production of gut microbiota metabolites [114].

The gut mucosal barrier acts as a physical and functional separation between the luminal content and the underlying compartment, which includes the gut epithelia, immune cells, blood vessels, and other structural elements in the lamina propria. Gut barrier dysfunction refers to the impaired integrity and function of the intestinal barrier [115]. Dysbiosis in the gut microbiota has been linked to heightened gut permeability in NAFLD [114]. The intricate relationship between gut dysbiosis and gut barrier dysfunction in NAFLD underscores the significance of gut microbiota in disease progression. It has been suggested that dysbiosis-mediated alterations in the structure and metabolism of gut microbes can affect the intestinal wall, leading to the migration of microbial products and consequent liver inflammation [5]. Certain gut microbial metabolites, trimethylamine N-oxide (TMAO), short-chain fatty acids (SCFAs), and lipopolysaccharides (LPS), are implicated in the occurrence and progression of NAFLD [116]. Trimethylamine N-oxide (TMAO) and lipopolysaccharides (LPS) promote hepatic lipid accumulation and inflammation [115,117,118]. SCFAs exhibit anti-inflammatory properties and improve insulin sensitivity [93]. Amino acids and bile acid metabolites impact inflammation, insulin resistance, and lipid metabolism [119,120,121]. Ethanol and gut dysbiosis are associated with oxidative stress in NAFLD [122,123,124].

#### 3.2.1. Trimethylamine N-oxide (TMAO)

Trimethylamine N-oxide (TMAO) is a gut microbial metabolite derived from dietary trimethylamine (TMA), primarily obtained from choline, phosphatidylcholine, and L-carnitine-rich foods. TMAO has gained attention due to its association with cardiovascular disease and metabolic disorders, including NAFLD. TMAO has been shown to promote hepatic lipid accumulation, inflammation, and fibrosis in animal models and human studies [117]. The ability of the gut microbiota to produce TMAO and its impact on lipid metabolism underscores the intricate relationship between gut microbial metabolites and the pathogenesis of NAFLD. Bacteria from various phyla, including Proteobacteria, Actinobacteria, and Firmicutes, especially those belonging to the *Clostridium XIVa cluster* and *Eubacterium*, have been confirmed to be involved in the production and metabolism of trimethylamine (TMA) from choline and carnitine, but further research is needed to explore the full extent of bacterial species involved [125,126]. Increased liver lipid peroxidation, as evidenced by higher malondialdehyde (MDA) levels, was observed in response to a high TMAO diet, along with elevated hepatic non-esterified fatty acid (NEFA) levels, according to Hu et al.’s study [127]. These findings indicate disrupted lipid metabolism and notable liver peroxidation damage. Moreover, the high TMAO diet resulted in a decrease in antioxidant enzyme activities, including hepatic total superoxide dismutase (T-SOD) and glutathione peroxidase (GSH-Px) activities, suggesting impaired antioxidant defense mechanisms in the liver [127]. Similarly, multiple studies have demonstrated that high choline intake and L-carnitine consumption, acting as precursors of trimethylamine-N-oxide (TMAO), can lead to the synthesis of TMAO by gut microbiota. Furthermore, the consumption of a diet rich in TMAO can further contribute to elevated TMAO levels in the body. These conditions have been associated with liver injury, disrupted lipid metabolism, and diminished antioxidant enzyme activities, such as SOD and GSH-Px. However, various interventions, including phloretin, chlorogenic acid (CGA), and total saponins of Gynostemma pentaphyllum (TSGP), have shown potential in mitigating hepatotoxicity induced by oxidative stress and restoring antioxidant capacity, particularly SOD and GSH-Px [128,129,130,131].

TMAO stimulates the transcription factor FoxO1, which is associated with metabolic disease, in the liver by interacting with the endoplasmic reticulum PKR-like eukaryotic initiation factor 2α kinase PERK (EIF2AK3), serving as a receptor. This interaction occurs through binding at physiological concentrations and specifically activating the unfolded protein response pathway (UPR) [132]. PERK serves as an ER stress sensor in eukaryotic cells, initiating an adaptive program that shapes the destiny of stressed cells by detecting unfolded proteins. Furthermore, PERK triggers NLRP3 inflammasome activation via the NF-κB pathway [133,134]. During ER stress, PERK sensors boost TXNIP transcription, hinder TXNIP mRNA breakdown by reducing miR-17, and thus modulate NLRP3 inflammasome activation [135,136]. An ER under stress leads to the buildup of oxidants, inducing oxidative stress [137]. In a study conducted by X. Sun et al., it was revealed that TMAO can trigger oxidative stress and activate the ROS-TXNIP-NLRP3 (NOD-like receptor protein 3) inflammasome signaling pathway in human umbilical vein endothelial cells (HUVECs) [138]. TXNIP operates through the NF-κB signaling pathway to activate the NLRP3 inflammasome [139]. TMAO exposure led to the production of ROS, which caused the disassociation of TXNIP from Trx and its binding to NLRP3 (Figure 4). This mediated the assembly of the NLRP3 inflammasome with Apoptosis-associated speck-like protein containing a CARD (ASC) and procaspase-1, leading to caspase-1-mediated conversion of pro-IL-1β to the activated form IL-1β. Ultimately, this led to the release of inflammatory cytokines and endothelial dysfunction [138].

#### 3.2.2. Lipopolysaccharides (LPS)

A range of health disorders, including NAFLD, as well as cardiovascular diseases, have been linked to LPS-triggered oxidative stress. LPS can upregulate NADPH oxidase and increase oxidative stress, potentially contributing to the development and progression of these diseases. Alterations in gut microbiota have been indicated to be associated with NADPH oxidase activation and redox signaling, highlighting the gut microbiota’s function in oxidative stress-related diseases [140]. In the liver, LPS-induced oxidative stress has been linked to mitochondrial biogenesis, cell proliferation, and activation of pro-survival signaling pathways. Exposure to LPS results in oxidative stress and depletion of mitochondrial GSH, leading to mitochondrial damage. However, LPS also stimulates the expression of genes involved in mitochondrial biogenesis, which helps maintain mitochondrial function and energy production. The PI3K/Akt pathway plays a crucial role in LPS-induced mitochondrial biogenesis and cell proliferation in response to oxidative stress [141]. LPS can disrupt lipid metabolism and impair insulin signaling, further exacerbating liver damage and promoting the development of steatosis and steatohepatitis in NAFLD [142,143].

Recognition of LPS by Toll-like receptor 4 (TLR4) on different cell types initiates the activation of inflammatory signaling pathways [115,118]. TLR4 binds LPS and activates NF-κB, initiating inflammation [144]. In the liver, the binding of LPS to TLR4 in macrophages (Kupffer cells) leads to the release of tumor necrosis factor-alpha (TNF-α) and interleukin-6 (IL-6), affecting hepatic stellate cells (HSCs) (Figure 5). Inducible nitric oxide synthase (iNOS) and the nitric oxide (NO) pathway play a role in stimulating Kupffer cells to generate oxidants when exposed to apoptotic bodies released from damaged hepatocytes [145,146]. Furthermore, myofibroblasts generated from activated hepatic stellate cells can also produce oxidants by NADPH oxidase, promoting oxidative stress and hepatic inflammation [147,148,149,150].

LPS-induced apoptosis is counteracted by Trx through the inhibition of ASK1 activation, a mediator of oxidants-induced cell death. In this process, Trx exhibits its antioxidant effects primarily in the cytosol, while the mitochondrial-specific Trx-2 is responsible for regulating apoptosis signaling pathways, collectively providing crucial protection against cell death caused by oxidants in distinct cellular compartments [151]. Further, Trx inhibits fibrosis by blocking the proliferation of HSC rather than directly blocking the apoptosis of hepatocytes. In addition, Trx has an inhibitory effect on HSC proliferation stimulated by serum and PDGF, which may be linked to its interaction with other signaling molecules or the inhibition of the p38 signal pathway in HSC (Figure 5) [152]. Xi Wang et al. investigated the role of Trx-2 in sepsis-induced liver damage [153]. They found that Trx-2 overexpression reduced inflammatory cytokine production, decreased neutrophil infiltration, and protected against liver injury in response to LPS stimulation. Moreover, decreased myeloperoxidase (MPO) activity and reduced levels of liver enzymes ALT and AST were observed, indicating a protective effect as a result of Trx-2 overexpression, ameliorating liver injury [153].

Antioxidants, such as N-Acetylcysteine (NAC) and tocopherol (TOC), have been used to investigate the role of oxidants in TLR4 signaling. These antioxidants have shown inhibitory effects on LPS-induced cytokine production, activation of various signaling pathways (IKK, ERK1/2, Akt, p38), and the activation of IRAK-1 and IRAK-4, key proteins in TLR4 signaling [154]. In models of GalN/LPS-induced fulminant hepatic failure, the NLRP3 inflammasome, oxidative stress, and related signaling molecules have been found to play significant roles. The NLRP3 inflammasome activation and cytokine responses, including IL-1β, have been associated with liver injury. Oxidant production and TXNIP-NLRP3 interaction have been implicated in NLRP3 inflammasome activation. Conversely, by inhibiting the activation of the NLRP3 inflammasome, heme oxygenase 1 (HO-1), an endogenous Nrf2-regulated enzyme, has been shown to mitigate GalN/LPS-induced mortality and hepatocellular necrosis [155]. In models of GalN/LPS-induced fulminant hepatic failure, the NLRP3 inflammasome, oxidative stress, and related signaling molecules have been found to play significant roles. The NLRP3 inflammasome activation and cytokine responses, including IL-1β, have been associated with liver injury. Oxidant production and TXNIP-NLRP3 interaction have been implicated in NLRP3 inflammasome activation.

#### 3.2.3. Short-Chain Fatty Acids (SCFAs)

Acetate, propionate, and butyrate are the major microbial metabolites produced through the anaerobic breakdown of dietary fiber by gut bacteria. These SCFAs play crucial roles in various physiological processes and have been associated with numerous health benefits. In NAFLD, they serve as an energy source for intestinal cells, fortifying the gut barrier and preserving optimal gut permeability, thus limiting the migration of harmful substances into the bloodstream [156]. SCFAs also possess anti-inflammatory properties and can improve insulin sensitivity, which are crucial factors in the development and progression of NAFLD [93]. SCFAs regulate immune cell recruitment and inflammatory responses through various signaling pathways, including GPR43 and NF-κB. By reducing hepatic inflammatory responses and promoting metabolic homeostasis, SCFAs have the potential to alleviate hepatic steatosis and NASH associated with NAFLD [157,158,159,160,161]. Through an AMP-activated protein kinase (AMPK)-dependent mechanism, SCFAs play a role in influencing hepatic lipid metabolism [162]. By upregulating the AMPK pathway, acetate, particularly produced by gut microbiota, inhibits chylomicron secretion from enterocytes and promotes lipid oxidation [163]. Moreover, SCFAs, including acetate and others, have been demonstrated to increase energy expenditure and promote lipid oxidation in the liver [164,165,166,167]. SCFAs exert their impact on energy expenditure and lipid oxidation through multifaceted mechanisms. They activate AMP-activated protein kinase (AMPK), a sensor responding to low cellular energy levels, promoting glucose uptake, fatty acid oxidation, and mitochondrial biogenesis. Additionally, SCFAs, particularly butyrate, activate insulin receptors, enhancing glucose uptake and indirectly supporting lipid oxidation. Notably, SCFAs also regulate lipid metabolism by modulating gene expression related to gluconeogenesis, potentially redirecting energy towards lipid oxidation [145]. However, the role of SCFAs in metabolism is complex, as excessive amounts of SCFAs, particularly from the consumption of fructose, can contribute to hepatic lipogenesis and fat accumulation [168].

Clinical and animal studies have suggested a connection between elevated short-chain fatty acid (SCFA) concentrations and an increased Firmicutes to Bacteroidetes ratio in obese individuals [169]. Elevated propionate levels, a specific type of SCFA, are found in NAFLD and obese patients due to gut dysbiosis. This increase in propionate levels is thought to play a role in the progression of NAFLD by sustaining a state of low-grade inflammation in the intestines [170]. Propionate and acetate are types of SCFAs that have the potential to promote lipid accumulation and gluconeogenesis in the liver [171]. Research findings indicate that butyrate initiates the expression of glucagon-like peptide 1 (GLP-1) in HepG2 cells by activating histone deacetylase 2 (HDAC-2), while not involving GPR43 and GPR109a receptors. Moreover, the introduction of sodium butyrate through supplementation has demonstrated significant efficacy in preventing the progression from simple steatosis to steatohepatitis, achieved through the orchestration of multiple intricate mechanisms [172,173]. The human gut contains important butyrate-producing bacteria mainly belonging to the Firmicutes phylum, such as *C. leptum*, *E. rectale*, *F. prausnitzii*, and *Roseburia* spp. These anaerobic bacteria, along with *E. hallii, Anaerostipes* spp., actively contribute to butyrate synthesis [174,175]. *Bacteroidetes* and *Clostridia* groups also play a significant role as dominant butyrate producers [176]. Additionally, other bacterial species, including *Bifidobacterium* and *Akkermansia muciniphila*, also participate in the production of diverse SCFAs [175,177,178]. Probiotics, prebiotics, and synbiotics (PPS) have emerged as therapeutic strategies with the ability to alter microbiota composition and reinstate microbial balance (13).

Probiotics encompass non-pathogenic living microorganisms, prebiotics entail indigestible fiber compounds, and synbiotics combine probiotics and prebiotics. Upon consumption, these agents can effectively bring about deliberate adjustments to gastrointestinal microbiota composition and activity while concurrently enhancing the release of endogenous intestinal nutrient peptides [179]. Carpi RZ et al. conducted a systematic review of 13 randomized controlled trials, encompassing 947 subjects, to assess interventions’ impact on gut microbiota and markers of NAFLD and NASH [180]. The studies revealed several noteworthy findings: (1) probiotic supplementation resulted in significant reductions in total cholesterol (TC), Body Mass Index (BMI), Aspartate Transaminase (AST), Alanine Transaminase (ALT), and liver stiffness, along with microbiota changes; (2) a multi-probiotic mixture showed improvements in markers such as AST, Interleukin-6 (IL-6), Tumor Necrosis Factor-alpha (TNF-α), and liver stiffness in NAFLD patients with Type 2 Diabetes (DM2); (3) a synbiotic containing diverse bacterial strains led to significant reductions in body weight, triglycerides, intrahepatic fat fraction, IL-6, and TNF-α; (4) yogurt consumption demonstrated positive effects on fat mass, lipid accumulation, Homeostatic Model Assessment of Insulin Resistance (HOMA-IR), and liver enzymes in obese women with NAFLD; and (5) prebiotic supplementation, such as fructo-oligosaccharides and inulin, displayed beneficial effects on NAFLD parameters, despite some studies having limitations like small sample sizes, short intervention times, and a lack of liver biopsies [180]. While both probiotics and prebiotics show promise in improving NAFLD/NASH markers, the authors emphasize the need for further research with larger sample sizes, longer intervention durations, and standardized assessment methods to comprehensively understand their mechanisms and therapeutic potential.

#### 3.2.4. Amino Acids

The gut microbiota actively metabolizes amino acids, including tryptophan, phenylalanine, and branched-chain amino acids (BCAAs), resulting in the production of metabolites that can significantly impact liver function in the development and progression of NAFLD (Figure 6). Tryptophan, an essential amino acid, serves as a precursor for multiple metabolites involved in NAFLD [181]. Indole, derived from tryptophan, enhances intestinal integrity and reduces inflammation and lipogenesis [182,183]. Indole-3-acetic acid, another derivative of indole, decreases hepatic lipogenesis inflammation and improves insulin resistance [184]. Serotonin, a neurotransmitter derived from tryptophan, when regulated by Thp1 (tryptophan hydroxylase 1, an enzyme in serotonin synthesis) and interacting with the HTR2a receptor (serotonin receptor 2a in the liver), plays a role in NAFLD by suppressing the energy expenditure of brown adipose tissue through inhibiting mitochondrial uncoupling protein 1 (UCP1) and promoting the accumulation of fat in the liver, known as hepatic steatosis [185,186]. A study done by Nocito et al. revealed that serotonin accumulation occurs in the liver primarily due to increased uptake by hepatic cells [119]. The study suggests that increased hepatic uptake of extracellular serotonin, possibly due to up-regulation of serotonin transporter (SERT), leads to its degradation by the enzyme MAO-A, generating oxidants that contribute to oxidative stress and liver damage in NASH. Oxidants are generated by the accumulated serotonin, which activates NADPH oxidase within HSCs. The increased levels of oxidants contribute to oxidative stress, which in turn impairs the normal functioning of mitochondria in HSCs. Mitochondria, the cellular powerhouses responsible for energy production, are particularly susceptible to oxidative damage [119]. There is not much research about Trx’s relation to serotonin in the liver context. However, levels and activity of the Trx in hippocampal cells were reported to be increased by chronic treatment with the antidepressants fluoxetine and venlafaxine. These drugs also inhibited protein cysteine sulfenylation induced by H_2_O_2_ and nitrosylation induced by the nitric oxide donor nitrosoglutathione. These findings suggest that Trx upregulation may potentially contribute to the protective effects of SSRIs and SNRIs against oxidative stress in depression [187]. Additional investigation is necessary to explore and modulate the potential correlation between Trx and serotonin in the context of the liver. The kynurenine pathway, also originating from tryptophan, is overactivated in NAFLD, leading to inflammation [182,184,185,188]. Phenylalanine and its derivative phenylacetic acid have been linked to hepatic steatosis by contributing to triglyceride accumulation and inhibiting insulin functions [120]. Furthermore, branched-chain amino acids (BCAAs) impair the tricarboxylic acid cycle and mitochondrial function, promoting insulin resistance and exacerbating NAFLD severity [189,190,191].

#### 3.2.5. Bile Acids Metabolites

Bile acids are essential for breaking down and absorbing dietary fats, and they are produced in the liver from cholesterol. In the metabolic pathway of bile acids, the gut microbiota exerts a significant impact by converting primary bile acids into secondary bile acids through enzymatic processes. This transformation takes place in the intestine, influencing the overall balance of bile acid metabolism in the body. These alterations in bile acid profiles can disrupt lipid metabolism, promote hepatic inflammation, and contribute to the development of hepatic steatosis and NAFLD [121]. Bile acids act mainly through farnesoid X receptor (FXR)- or Takeda G-protein-coupled bile acid protein 5 (TGR5)-mediated signaling pathways. FXR, a nuclear receptor, has multiple effects on lipid and glucose metabolism. Under the influence of FXR activation, several metabolic processes are affected. It leads to the inhibition of gluconeogenesis-related gene expression, promotes liver glycogen synthesis, inhibits lipogenesis, promotes the oxidation of fatty acids, and influences cholesterol transport. Furthermore, FXR stimulates the production of liver fibroblast growth factor 21 (FGF21), which facilitates glucose uptake by adipose tissue and enhances insulin sensitivity [192,193,194,195]. On the other hand, bile acids activate the TGR5 receptor in various cell types involved in inflammation regulation, such as nonparenchymal hepatocytes, monocytes, and macrophages. This activation leads to the suppression of inflammatory mediators like interleukin-6 (IL-6), IL-1A, and IL-1B. Moreover, through a TGR5-cAMP-dependent pathway, bile acids inhibit the secretion of tumor necrosis factor by Kupffer cells [196,197].

Na Jiao et al. conducted a study that revealed that NAFLD was linked to elevated levels of both primary and secondary bile acids in the bloodstream, resulting from the activity of the gut microbiota [198]. Specifically, the study revealed elevated concentrations of deoxycholic acid (DCA), which acts as an antagonist of FXR, and decreased levels of the agonistic bile acid chenodeoxycholic acid (CDCA) in NAFLD. In a study conducted by Masahiro Nomoto et al. [164], it was observed that *Fxr*-null mice exhibited significantly higher levels of oxidative stress markers, including 8-hydroxy-2′-deoxyguanosine (8OHdG), thiobarbituric acid-reactive substances (TBARS), and hydroperoxides, compared to wild-type mice. Furthermore, the expression of oxidative stress-related genes and the Nrf2 were upregulated in the livers of *Fxr*-null mice [199]. Also, Wang et al. discovered that FXR has the ability to inhibit NF-κB [200]. Thus, as discovered by Zhizhen Xu et al., it was found that administration of the FXR natural ligand, CDCA, in an animal model of LPS-induced liver injury could alleviate inflammatory damage to hepatocytes [201]. It has been reported that NF-κB may upregulate the expression or activity of two crucial cellular antioxidants, Trx1 and Trx2 [202,203]. Taken together, increased levels of DCA, an antagonist of FXR, and decreased levels of the agonisticCDCA in NAFLD can lead to antagonizing FXR signaling and activation of NF-κB, potentially impacting the thioredoxin antioxidant system (Figure 7).

#### 3.2.6. Ethanol

Ethanol levels are linked to alterations in the gut microbiota, indicating a potential role of dysbiosis in NAFLD development, with specific bacteria like Escherichia, Bacteroides, Bifidobacterium, Clostridium, and Klebsiella pneumonia associated with alcohol production and its relation to NAFLD [122,123,124]. Notably, increased levels of ethanol have been observed in NASH patients, particularly in those with higher quantities of alcohol-producing Escherichia bacteria [122,123]. Further investigations have demonstrated that the introduction of *Klebsiella pneumonia* strains that produce elevated alcohol levels in mice can induce NAFLD-like changes [124]. Furthermore, abnormal ethanol metabolism and heightened blood–ethanol levels have been linked to insulin resistance in children diagnosed with NAFLD [204]. Alcohol metabolism by enzymes like alcohol dehydrogenase (ADH) and cytochrome P4502E1 (CYP2E1) generates oxidants. Oxidants induce cellular damage by oxidizing lipids, inhibiting mitochondrial function, and disrupting fatty acid oxidation, ultimately leading to the accumulation of intracellular lipids and the development of hepatic steatosis [205]. Additionally, alcohol-induced oxidant production can lead to mitochondrial dysfunction, impaired mitochondrial protein synthesis, and release of proapoptotic factors. Furthermore, alcohol metabolism produces reactive aldehydes that can form adducts with proteins, contributing to inflammation and liver injury [205]. Moreover, Trx-1 protein levels in the liver are depleted by ethanol exposure, potentially heightening the liver’s susceptibility to injury. This reduction in Trx-1 protein is not linked to changes in Trx-1 mRNA levels, indicating that its regulation might occur at the posttranscriptional level [206]. While the precise roles of ethanol in different types of NAFLD remain a topic of debate, additional research is necessary to unravel its mechanisms and understand its significance.

## 4. Therapeutical Potential for NAFLD/NASH with Specifically Thiol Redox Regulation

Based on the role of thiol-redox systems in the pathogenesis of NAFLD/NASH, regulation of cellular redox homeostasis emerges as a new pharmacotherapeutic strategy for NAFLD/NASH treatment. Prominent examples of these medications include N-acetylcysteine, 2,3-meso-dimercaptosuccinic acid, British anti-Lewisite, D-penicillamine, and amifostine [207]. These pharmaceuticals that contain thiol groups exhibit the capability to reduce the presence of harmful free radicals and other noxious electrophilic substances. These compounds have the potential to replenish cellular thiol reserves and form long-lasting complexes with heavy metals such as lead, arsenic, and copper. Consequently, thiols demonstrate their adaptability in treating a wide range of conditions by acting as scavengers of radicals, precursors for GSH, and agents for binding metals. Initial research suggests that taking GSH orally (300 mg/day) for four months can reduce ALT levels and hepatic steatosis in Japanese NAFLD patients without severe fibrosis or uncontrolled diabetes. However, further extensive clinical trials are necessary to confirm its effectiveness [208]. NAC has increasingly become the prime example of an “antioxidant.” It is arguable that it holds the position as the most extensively employed “antioxidant” in both experimental cellular and animal biology, as well as clinical trials. Many researchers utilize and assess NAC with the notion that it has the potential to hinder or reduce oxidative stress. Traditionally, it is presumed that NAC functions as (i) a reducer of disulfide bonds, (ii) a scavenger of reactive oxygen species, and/or (iii) a precursor for the production of glutathione. Although these mechanisms might be applicable in specific scenarios, they cannot be broadly applied to elucidate the impacts of NAC across a majority of contexts and conditions [209]. Supplementing with NAC greatly enhanced the effects of high-fat diet-induced obesity, abnormal lipid levels, and liver issues in mice. NAC also reversed the disruption of gut microbiota caused by the high-fat diet. Interestingly, the positive impact of NAC supplementation on reducing liver fat and damage was eliminated when intestinal microorganisms were removed using antibiotics, suggesting that the gut microbiota plays a crucial role in NAC’s beneficial effect [210].

Apart from the direct regulation of thiol-redox systems, some other antioxidants exhibit therapeutical potential for NAFLD/NASH. For example, staxanthin lessens weight gain and adipose tissue growth and enhances lipid metabolism, decreasing liver weight, triglycerides, plasma triglycerides, and total cholesterol. It possesses anti-inflammatory properties, reducing inflammatory macrophages, curbing immune cell recruitment in the liver, and countering inflammation in non-alcoholic fatty liver disease. It also bolsters insulin signaling, lowers lipid accumulation, and inhibits pro-inflammatory signals. Moreover, astaxanthin spurs autophagy in liver cells, breaking down stored lipid droplets and influencing pathways that mitigate inflammation and oxidative stress. Notably, it also impacts gene expression related to these processes [211].

## 5. Conclusions

The pathogenesis of NAFLD involves several mechanisms that disrupt hepatic lipid homeostasis, with the Trx/Grx system playing a crucial role in disease development. Maintaining a balanced redox state is essential for normal cellular physiology, and excessive lipid accumulation in liver cells can lead to an oversaturation of free radicals, causing disturbances in lipid metabolism and subsequent oxidative damage, mitochondrial dysfunction, and endoplasmic reticulum stress in liver cells. This further exacerbates liver inflammation and hepatocyte fibrosis. Therefore, regulating the body’s redox balance is vital for maintaining hepatic lipid homeostasis.

The development and progression of NAFLD are significantly influenced by the interplay between gut microbiota dysbiosis and thiol-redox system-mediated oxidative stress. By inducing dysbiosis, the gut barrier becomes disrupted, resulting in heightened permeability and the migration of harmful microbial products into the liver, consequently triggering inflammation and oxidative stress. The gut microbiota metabolites, such as TMAO, LPS, and SCFAs, contribute to liver inflammation, lipid accumulation, and insulin resistance. Elevated levels of TMAO and LPS contribute to oxidative stress, while SCFAs have anti-inflammatory and metabolic benefits in NAFLD. Additionally, ethanol and specific bacteria associated with alcohol production are linked to alterations in the gut microbiota, leading to oxidative stress and insulin resistance in NAFLD. Understanding the intricate relationship between gut microbiota dysbiosis and thiol-redox systems in NAFLD opens avenues for potential therapeutic strategies. Targeting both factors through interventions such as probiotics, prebiotics, dietary modifications, and antioxidant therapies may hold promise in preventing and treating NAFLD. Further research is necessary to unravel the precise mechanisms involved and identify novel therapeutic approaches to effectively manage this prevalent liver disease. By addressing both gut microbiota dysbiosis and oxidative stress, we can potentially mitigate the development and progression of NAFLD and improve patient outcomes.

## Figures and Tables

**Figure 1 antioxidants-12-01680-f001:**
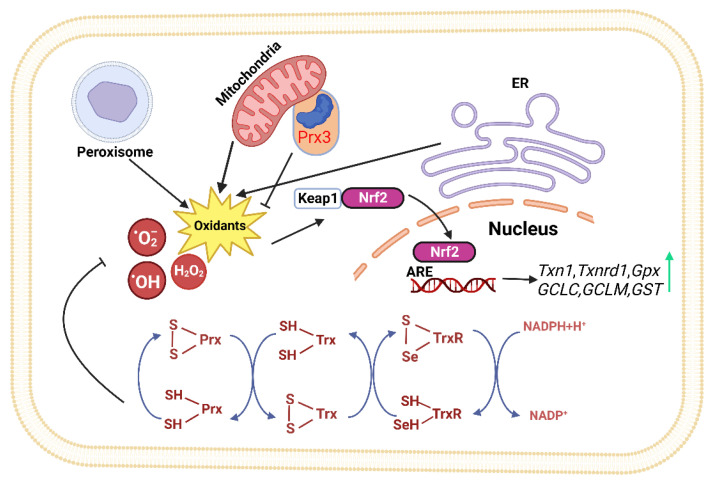
The thioredoxin system plays a crucial role in modulating cell viability and proliferation. Thioredoxin can donate electrons to various enzymes, including peroxiredoxins, which have critical roles in cell signaling by either removing hydrogen peroxide or regulating redox-sensitive signaling molecules [40,41,42,43,44,45]. The redox state of thioredoxin can affect the function of several transcription factors, making it an important player in cellular signaling [46,47,48,49]. (ER: Endoplasmic reticulum, Keap1: kelch like ECH associated protein 1, Nrf2: Nuclear factor erythroid 2-related factor 2, ARE: Antioxidant response element, TrxR: Thioredoxin Reductase, Trx: Thioredoxin, Prx: Peroxiredoxin, NADPH: Nicotinamide Adenine Dinucleotide Phosphate).

**Figure 2 antioxidants-12-01680-f002:**
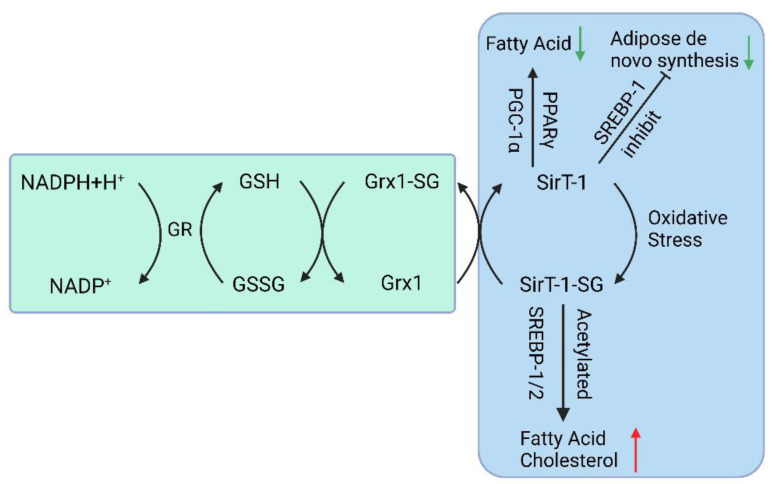
The GSH-Grx 1 system controls protein *S*-glutathionylation major players in the NAFLD. (GR: Glutathione Reductase, GSH: Glutathione, NADPH: Nicotinamide Adenine Dinucleotide Phosphate, Grx: Glutaredoxin, SirT-1: sirtuin type 1, PPAR: Peroxisome proliferators-activated receptors, PGC-1α: Peroxisome proliferator-activated receptor-gamma coactivator, SREBP-1c: Sterol regulatory element binding protein-1c).

**Figure 3 antioxidants-12-01680-f003:**
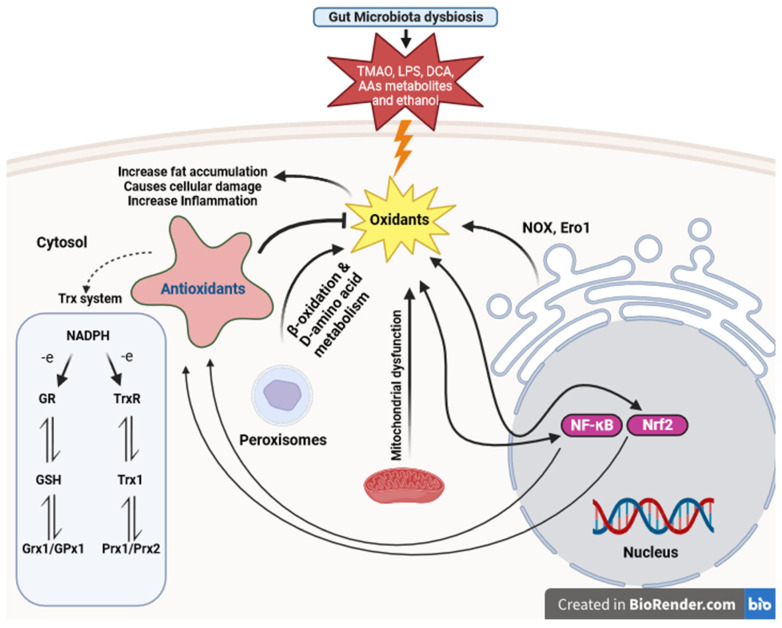
Thioredoxin and glutathione system participants in defense against oxidative stress caused by gut microbiota dysbiosis in NAFLD. This figure illustrates the dynamic interplay among gut microbiota dysbiosis, oxidative stress, mitochondrial dysfunction, endoplasmic reticulum stress, inflammation, and peroxisome-related processes in the pathogenesis of NAFLD. (LPS: lipopolysaccharides, TMAO: trimethylamine N-oxide, AAs: Amino acids, DCA: deoxycholic acid, NF-κB: Nuclear Factor-kappa B, Nrf2: Nuclear factor erythroid 2-related factor 2, ERO1: Endoplasmic Reticulum Oxidoreductin 1, NOX: NADPH Oxidase, GR: Glutathione Reductase, GSH: Glutathione, TrxR: Thioredoxin Reductase, Trx: Thioredoxin, NADPH: Nicotinamide Adenine Dinucleotide Phosphate, Grx1: Glutaredoxin 1, GPx1: Glutathione Peroxidase 1, Prx1: Peroxiredoxin 1, Prx2: Peroxiredoxin 2).

**Figure 4 antioxidants-12-01680-f004:**
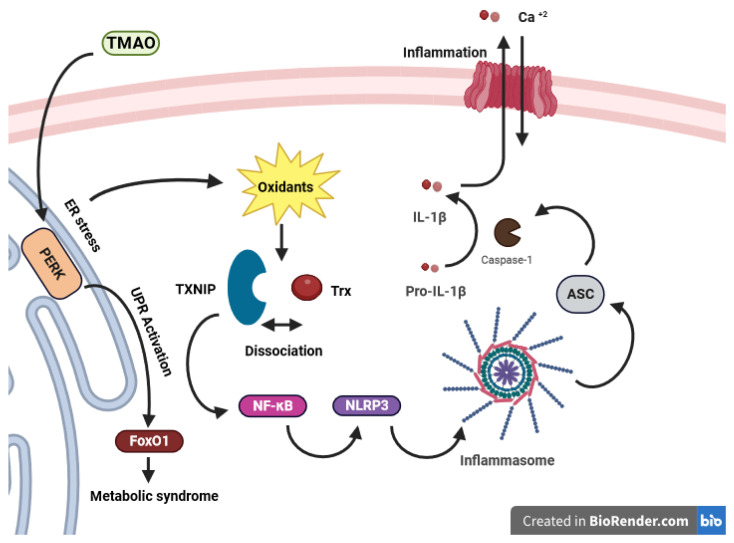
Trx regulation in TMAO-caused inflammation. TMAO induces the activation of FoxO1 in the liver by binding to PERK, an ER stress sensor, which initiates the unfolded protein response (UPR). This process not only triggers oxidative stress but also contributes to the development of metabolic syndrome. This, in turn, triggers the NLRP3 inflammasome through the NF-κB pathway. This activation of NF-κB may stem from oxidant release due to ER-mediated stress, leading to the dissociation of TXNIP from Trx and subsequent binding and activation of NF-κB. NF-κB then prompts the induction of NLRP3, leading to the assembly with ASC and procaspase-1. This leads to caspase-1-mediated conversion of pro-IL-1β to the activated form IL-1β and triggering the inflammation. TMAO exposure causes oxidative stress and inflammatory cytokine release in endothelial cells, contributing to metabolic disease. (TMAO: trimethylamine N-oxide, PERK: PKR-like eukaryotic initiation factor 2α kinase, UPR: Unfolded Protein Response, NLRP3: NOD-like receptor protein 3, NF-κB: Nuclear Factor-Kappa B, TXNIP: Thioredoxin-Interacting Protein, ASC: Apoptosis-associated speck-like protein containing a CARD).

**Figure 5 antioxidants-12-01680-f005:**
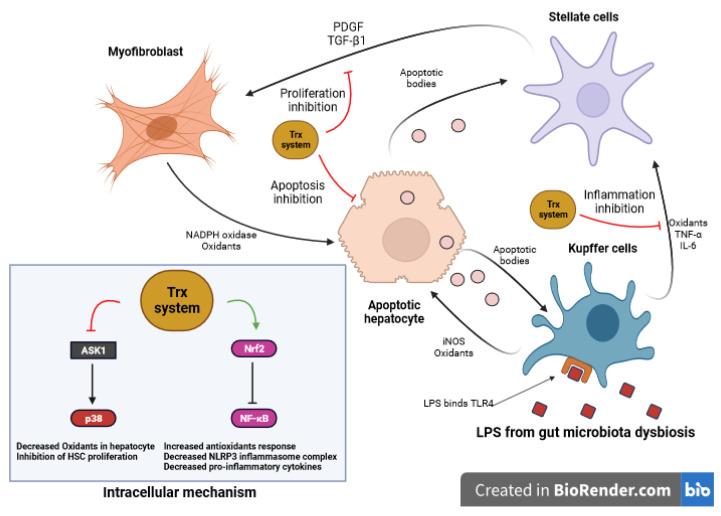
Role of Trx system in LPS impact on liver cells. LPS initiates inflammation by activating TLR4 on various cells. In the liver, LPS binding to TLR4 in macrophages (Kupffer cells) leads to the release of TNF-α and IL-6, affecting HSCs. Concurrently, iNOS induces the generation of oxidants in Kupffer cells, while activated HSCs produce oxidants through NADPH oxidase, resulting in oxidative stress and inflammation. The antioxidant Trx counteracts LPS-induced apoptosis by inhibiting the activation of ASK1, with Trx-2 located in mitochondria, regulating apoptosis signaling and collectively protecting against cell death from oxidants. Additionally, Trx inhibits HSC proliferation, reduces fibrosis, and interacts with inflammatory pathways, offering protective effects. Trx-2 overexpression reduces inflammation, neutrophil infiltration, and liver injury caused by LPS exposure, contributing to hepatic health. (LPS: Lipopolysaccharide, TLR4: Toll-like receptor 4, TNF-α: Tumor Necrosis Factor-Alpha, IL-6: Interleukin-6, iNOS: Inducible Nitric Oxide Synthase, NADPH: Nicotinamide Adenine Dinucleotide Phosphate, ASK1: Apoptosis Signal-Regulating Kinase 1, Trx: Thioredoxin, Trx-2: Mitochondrial Thioredoxin)”.

**Figure 6 antioxidants-12-01680-f006:**
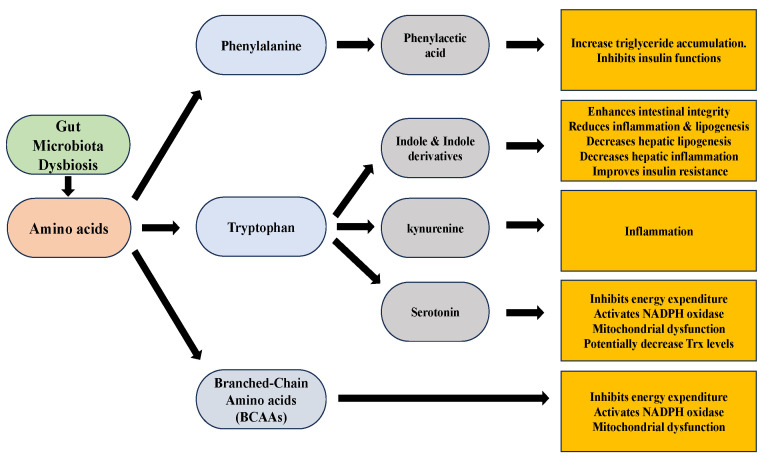
Amino acids metabolites and their effects.

**Figure 7 antioxidants-12-01680-f007:**
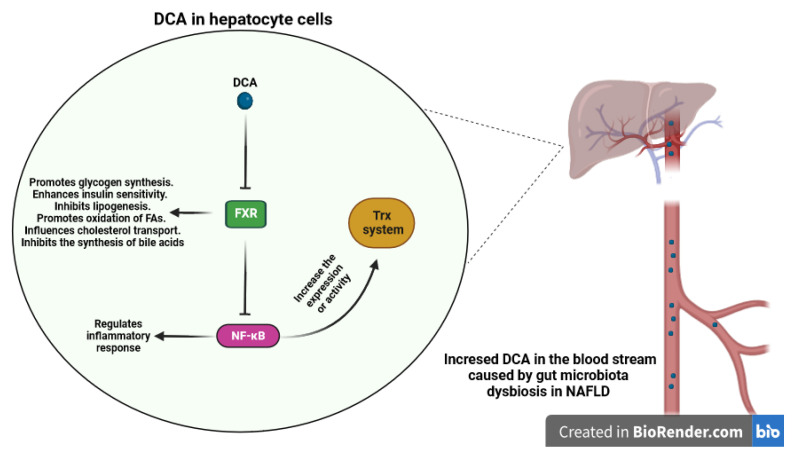
Impact of DCA on the hepatocytes and antioxidant system. Elevated concentrations of DCA, acting as an antagonist of FXR in NAFLD, have the potential to antagonize FXR signaling. This antagonistic effect may lead to the activation of NF-κB, a transcription factor central to inflammation. This intriguing interaction could potentially influence the function of the thioredoxin antioxidant system. (DCA: Deoxycholic Acid, FXR: Farnesoid X Receptor, NF-κB: Nuclear Factor-Kappa B).

## Data Availability

All of the data are contained within the article.

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
