# Peer review of "Thioredoxin/Glutaredoxin Systems and Gut Microbiota in NAFLD: Interplay, Mechanism, and Therapeutical Potential"

_antioxidants, 2023, doi:10.3390/antiox12091680_

Round 1

Reviewer 1 Report

In this manuscript, Zhu and collaborators made a tentative review regarding the interplay and mechanistic role of Thioredoxin/glutaredoxin systems and gut microbiota in NAFLD. The review is well written, although it is based on main correlations found in gut microbiota and metabolites and those changes reported in the thioredoxin/glutaredoxin system in the liver, with some lack of facts. It is unclear whether the microbiota changes could be extrapolated to the liver metabolism. A change in the title is suggested, and additional discussion is suggested to be included regarding the following points to improve the quality of the review.

Major comments:

1) The authors indicated that Thioredoxin/glutaredoxin systems are mainly regulated by Nfr-2, but this is a very reductionist hypothesis since these systems are regulated by many elements (see: Biochim Biophys Acta . 2014;1840(1):303-14.). Please, include a section regarding modulation of the gene expression of thioredoxin/glutaredoxin systems by alternative elements to Nrf-2 that might be involved in NAFLD/NASH.

2) Point 3.2 of the review should be reformulated or eliminated. As indicated in lines 271-283, individuals with NAFLD and NASH present intestinal wall damage, promoting microbial product migration and consequent liver inflammation. This is an incorrect assumption with no evidence. A loss of function in the intestinal barrier, such as that accounting for inflammatory bowel disease, does not account for NAFLD/NASH. Indeed, the existence of an intact intestinal barrier is compatible with the permeabilization of some microbe’s components/metabolites toward the blood barrier, as those commented in the next section of the review 3.3

3) Based on Figure 4, the target of TMAO is located at the plasma membrane leading to ROS formation. The authors should clarify if TMAO exposure's effects are directly or indirectly associated with ROS. Other possible targets different from those present in the plasma membrane should be included in Figure 4, as indicated in the manuscript. Please modify the figure.

4) Authors commented on the SCFAs protecting role against NASH and NAFLD. At this point, authors should indicate the effect of gut dysbiosis on SCFAs production and whether prebiotics and probiotics might improve NASH and NAFLD.

6) A section regarding the potential use of antioxidants containing SH groups might affect NASH and NAFLD.

7)) Please specify the mechanism of action for these changes:

Regarding an increase:

a) Line 149-152: “A diet rich in high-saturated fatty acids (HSFA) significantly increased the mRNA levels of Trx in adipose tissue. Simultaneously, postprandial adipose tissue TrxR1 mRNA significantly decreased, indicating increased oxidative stress due to saturated fat intake” Why?

b) Line 153-154: “In obese individuals with metabolic disorders, subcutaneous tissue TrxR activity and Trx content are significantly increased” Why?

c) line 180-181: “S-glutathionylated protein levels increase, and an important target protein, Sirtuin-1 (SirT1), is identified.” Why?

d) Line 246-247: “Higher Trx was found to be positively associated with the severity of NASH and increased iron accumulation in the liver” Why?

e) Line 249-250: “NASH induction led to increased expression of the TXNIP gene and reduced expression of TrxR1 and TrxR2 in the liver” Why?

f) Line 405-406: “SCFAs, including acetate and others, have been demonstrated to increase energy expenditure and promote lipid oxidation in the liver” Why?

g) Line 424-426: “Increased serotonin levels derived from tryptophan inhibit energy expenditure and contribute to hepatic steatosis” Why?

h) Line 427-428 “ A study done by Nocito et al. revealed that serotonin accumulation occurs in the liver, primarily due to increased uptake by HSCs “ Why?

Minor comments:

Line 325, the word ASC is colored in red for no reason.

Please define ASC, NAC, and TOC, h the first time named in the manuscript.

Author Response

In this manuscript, Zhu and collaborators made a tentative review regarding the interplay and mechanistic role of Thioredoxin/glutaredoxin systems and gut microbiota in NAFLD. The review is well written, although it is based on main correlations found in gut microbiota and metabolites and those changes reported in the thioredoxin/glutaredoxin system in the liver, with some lack of facts. It is unclear whether the microbiota changes could be extrapolated to the liver metabolism. A change in the title is suggested, and additional discussion is suggested to be included regarding the following points to improve the quality of the review.

>>> Thank the reviewer for this comment. Your insights and suggestions have significantly improved the quality and clarity of our work. We have meticulously addressed each of the major and minor comments provided by the reviewer, making the necessary adjustments to enhance the manuscript's content and presentation. We are genuinely grateful for the thorough review, as it has greatly contributed to creating a more comprehensive and well-structured document. If there are any additional aspects that require adjustment or specific areas you would like us to focus on, please let us know. Your guidance is crucial in ensuring that the manuscript meets the highest standards. We are dedicated to refining the document to your satisfaction and meeting all necessary requirements. We have added one part for the therapeutical potential, thus we have changed the title.

Major comments:

1) The authors indicated that Thioredoxin/glutaredoxin systems are mainly regulated by Nfr-2, but this is a very reductionist hypothesis since these systems are regulated by many elements (see: Biochim Biophys Acta . 2014;1840(1):303-14.). Please, include a section regarding modulation of the gene expression of thioredoxin/glutaredoxin systems by alternative elements to Nrf-2 that might be involved in NAFLD/NASH.

>>>Thank the reviewer for pointing out this important concern. Nrf2 signaling pathway, as a major antioxidant pathway, plays a very important role in combating oxidative stress. When NAFLD undergoes oxidative stress and produces excess oxidants, Nrf2 binds to the antioxidant progenitor ARE to regulate target genes, which include Txn, Txnrd, in addition, specificity protein 1(Sp1), Fas/ Jun, TATA box binding protein (TBP), cAMP response element binding protein (CREB), retinoic acid receptor/Retinoid X receptor (RAR/RXR), and other transcription factors bind to their DNA sites to regulate Txn; transcription factors such as Octamer binding protein (Oct-1), Sp1, Sp3, and other transcription factors bind to their DNA sites to regulate Txnrd; however, these do not appear to have a strong association with oxidative stress, and it is unclear whether these transcriptional regulators are present in NAFLD/NASH. We have added a section to describe these contents.

2) Point 3.2 of the review should be reformulated or eliminated. As indicated in lines 271-283, individuals with NAFLD and NASH present intestinal wall damage, promoting microbial product migration and consequent liver inflammation. This is an incorrect assumption with no evidence. A loss of function in the intestinal barrier, such as that accounting for inflammatory bowel disease, does not account for NAFLD/NASH. Indeed, the existence of an intact intestinal barrier is compatible with the permeabilization of some microbe’s components/metabolites toward the blood barrier, as those commented in the next section of the review 3.3.

>>>Thank the reviewer for this suggestion. We have reformulated this section. Point 3.3 of the review presents an assertion that individuals with NAFLD and NASH have intestinal wall damage, which promotes the migration of microbial products and subsequently leads to liver inflammation. However, this assertion lacks supporting evidence and is an inaccurate assumption. Contrary to this idea, it's important to note that a loss of function in the intestinal barrier, as seen in conditions like inflammatory bowel disease, does not directly account for the development of NAFLD or NASH. In fact, it is possible for the intestinal barrier to remain intact while still allowing the permeabilization of certain microbial components and metabolites into the bloodstream, as will be discussed further in the subsequent section of the review (3.1.1.).

3) Based on Figure 4, the target of TMAO is located at the plasma membrane leading to ROS formation. The authors should clarify if TMAO exposure's effects are directly or indirectly associated with ROS. Other possible targets different from those present in the plasma membrane should be included in Figure 4, as indicated in the manuscript. Please modify the figure.

>>> We have modified the figure 4. The updated Figure 4 presents an expanded view of TMAO's impact on cellular pathways, including potential targets beyond the plasma membrane that contribute to ROS-mediated effects.

4) Authors commented on the SCFAs protecting role against NASH and NAFLD. At this point, authors should indicate the effect of gut dysbiosis on SCFAs production and whether prebiotics and probiotics might improve NASH and NAFLD.

>>> Thank you for this suggestion. We have incorporated the suggested information regarding the effect of gut dysbiosis on SCFAs production and the potential role of prebiotics and probiotics in improving NASH and NAFLD. The following information were added to the text:

Clinical and animal studies have suggested a connection between elevated short-chain fatty acid (SCFA) concentrations and an increased Firmicutes to Bacteroidetes ratio in obese individuals [175]. Elevated propionate levels, a specific type of SCFA, are found in NAFLD and obese patients due to gut dysbiosis. This increase in propionate levels is thought to play a role in the progression of NAFLD by sustaining a state of low-grade inflammation in the intestines [176]. Propionate and acetate are types of short-chain fatty acids (SCFAs) that have the potential to promote lipid accumulation and gluconeogenesis in the liver [177]. Research findings indicate that butyrate initiates the expression of glucagon-like peptide 1 (GLP-1) in HepG2 cells by activating histone deacetylase 2 (HDAC-2), while not involving GPR43 and GPR109a receptors. Moreover, the introduction of sodium butyrate through supplementation has demonstrated significant efficacy in preventing the progression from simple steatosis to steatohepatitis, achieved through the orchestration of multiple intricate mechanisms.

Probiotics, prebiotics, and synbiotics (PPS) have emerged as therapeutic strategies with the ability to alter microbiota composition and reinstate microbial balance (13). Probiotics encompass non-pathogenic living microorganisms, prebiotics entail indigestible fiber compounds, and synbiotics combine probiotics and prebiotics. Upon consumption, these agents can effectively bring about deliberate adjustments to gastrointestinal microbiota composition and activity, while concurrently enhancing the release of endogenous intestinal nutrient peptides [185]. Carpi RZ et al. conducted a systematic review of 13 randomized controlled trials, encompassing 947 subjects, to assess interventions' impact on gut microbiota and markers of NAFLD and NASH [186]. The studies revealed several noteworthy findings: (1) probiotic supplementation resulted in significant reductions in Total Cholesterol (TC), Body Mass Index (BMI), Aspartate Transaminase (AST), Alanine Transaminase (ALT), and liver stiffness, along with microbiota changes; (2) a multi-probiotic mixture showed improvements in markers such as AST, Interleukin-6 (IL-6), Tumor Necrosis Factor-alpha (TNF-α), and liver stiffness in NAFLD patients with Type 2 Diabetes (DM2); (3) a synbiotic containing diverse bacterial strains led to significant reductions in body weight, triglycerides, intrahepatic fat fraction, IL-6, and TNF-α; (4) yogurt consumption demonstrated positive effects on fat mass, lipid accumulation, Homeostatic Model Assessment of Insulin Resistance (HOMA-IR), and liver enzymes in obese women with NAFLD; and (5) prebiotic supplementation, such as fructo-oligosaccharides and inulin, displayed beneficial effects on NAFLD parameters, despite some studies having limitations like small sample sizes, short intervention times, and a lack of liver biopsies [186]. While both probiotics and prebiotics show promise in improving NAFLD/NASH markers, the authors emphasize the need for further research with larger sample sizes, longer intervention durations, and standardized assessment methods to comprehensively understand their mechanisms and therapeutic potential.

6) A section regarding the potential use of antioxidants containing SH groups might affect NASH and NAFLD.

>>> Thank you for this excellent suggestion. We have added one section concerning the use of SH-containing antioxidant in NAFLD (section 4).

7)) Please specify the mechanism of action for these changes:

Regarding an increase:

  1. a) Line 149-152: “A diet rich in high-saturated fatty acids (HSFA) significantly increased the mRNA levels of Trx in adipose tissue. Simultaneously, postprandial adipose tissue TrxR1 mRNA significantly decreased, indicating increased oxidative stress due to saturated fat intake” Why?

>>> Trx acts as an antioxidant protein involved in the reduction of other proteins via cysteine thiol-disulfide exchanges and also acts as an electron donor for peroxides. Relative to the high-monounsaturated fatty acid diet (HMUFA) and the low-fat, high-complex carbohydrate diet (LFHCC), diets enriched in highly saturated fatty acids (HSFA) significantly increased mRNA levels of Trx in adipose tissue, and also Adipose tissue levels of TrxR1 mRNA, were significantly lower, suggesting that the ability of Trx to revert from its oxidized form to its reduced form is reduced, leading to a compensatory increase in Trx gene levels. This indicate that regulation of Trx and TrxR expression is not always consistent under certain condition. The expression of Trx and TrxR may be predominantly controlled by different transcription factors. This is a very interesting question. But the detail mechanism behind is still not clear.

  1. b) Line 153-154: “In obese individuals with metabolic disorders, subcutaneous tissue TrxR activity and Trx content are significantly increased” Why?

>>>Because subcutaneous fat seems to provide good protection against increased oxidation in obese subjects compared to visceral fat, TrxR and Trx content are significantly increase for the compensation effects. In addition, protein levels of Trx, GPx, and CuZnSOD, as well as activities of GPx, GR, GST, TrxR, and SOD, were significantly higher in "at-risk" obese women than in metabolically healthy women.

  1. c) line 180-181: “S-glutathionylated protein levels increase, and an important target protein, Sirtuin-1 (SirT1), is identified.” Why?

>>> We have added more description about S-glutathionylation of Sirtuin-1 in the text as follows:  “SirT1 is a NAD+-dependent class III histone deacetylase that regulates key transcription factors coordinating hepatic lipid metabolism. Activation of SirT1 ameliorates NAFL; conversely, hepatic SirT1 deficiency leads to steatosis, and thiol modification of SirT1 regulates lipid metabolism through acetylation of key transcription factors. S-glutathionylation inactivates SirT1 and promotes hyperacetylation and activation of downstream target proteins such as p53 and sterol regulatory element binding protein (SREBP).”

  1. d) Line 246-247: “Higher Trx was found to be positively associated with the severity of NASH and increased iron accumulation in the liver” Why?

>>> The observed increase in iron accumulation in the liver may potentially be attributed to a phenomenon known as ferroptosis. Ferroptosis is an iron- and thiol-associated oxidative stress process that has been implicated in influencing the pathogenesis of NASH. This process involves intricate cellular mechanisms that warrant further investigation to better understand its role in contributing to the iron buildup within the liver. 

  1. e) Line 249-250: “NASH induction led to increased expression of the TXNIP gene and reduced expression of TrxR1 and TrxR2 in the liver” Why?

>>> Oxidative stress triggers the upregulation of TXNIP expression, by the activation of its promoter containing a carbohydrate response element (ChoRE), which is modulated by transcription factors including MondoA:Max-like protein X (MLx), nuclear factor Y (NF-Y), and carbohydrate response element-binding protein (ChREBP). We have added the description in the text.

  1. f) Line 405-406: “SCFAs, including acetate and others, have been demonstrated to increase energy expenditure and promote lipid oxidation in the liver” Why?

>>>SCFAs exert their impact on energy expenditure and lipid oxidation through multifaceted mechanisms. They activate AMP-activated protein kinase (AMPK), a sensor responding to low cellular energy levels, promoting glucose uptake, fatty acid oxidation, and mitochondrial biogenesis. Additionally, SCFAs, particularly butyrate, activate insulin receptors, enhancing glucose uptake and indirectly supporting lipid oxidation. Notably, SCFAs also regulate lipid metabolism by modulating gene expression related to gluconeogenesis, potentially redirecting energy towards lipid oxidation. We have added the description in the text.

  1. g) Line 424-426: “Increased serotonin levels derived from tryptophan inhibit energy expenditure and contribute to hepatic steatosis” Why?

>>>Serotonin, a neurotransmitter derived from tryptophan, when regulated by Thp1 (tryptophan hydroxylase 1, an enzyme in serotonin synthesis) and interacting with the HTR2a receptor (serotonin receptor 2a in the liver), plays a role in non-alcoholic fatty liver disease (NAFLD) by suppressing the energy expenditure of brown adipose tissue through inhibiting mitochondrial uncoupling protein 1 (UCP1) and promoting the accumulation of fat in the liver, known as hepatic steatosis. A study done by Nocito et al. revealed that serotonin accumulation occurs in the liver, primarily due to increased uptake by hepatic cells [120]. The study suggests that increased hepatic uptake of extracellular serotonin, possibly due to up-regulation of serotonin transporter (SERT), leads to its degradation by the enzyme MAO-A, generating reactive oxygen species (ROS) that contribute to oxidative stress and liver damage in NASH.

  1. h) Line 427-428 “A study done by Nocito et al. revealed that serotonin accumulation occurs in the liver, primarily due to increased uptake by HSCs “Why?

>>>A study done by Nocito et al. revealed that serotonin accumulation occurs in the liver, primarily due to increased uptake by hepatic cells [120]. The study suggests that increased hepatic uptake of extracellular serotonin, possibly due to up-regulation of serotonin transporter (SERT), leads to its degradation by the enzyme MAO-A, generating reactive oxygen species (ROS) that contribute to oxidative stress and liver damage in NASH.

Minor comments:

Line 325, the word ASC is colored in red for no reason.

>>>The formatting issue regarding the word "ASC" in Line 325 has been corrected.

Please define ASC, NAC, and TOC, h the first time named in the manuscript.

>>> It has been revised and defined in the text.

Reviewer 2 Report

The manuscript of Zhu et al. aims to provide a comprehensive review on the topic of the interaction between the thioredoxin/glutaredoxin (Trx/Grx) systems, gut microbiota and non-alcoholic fatty liver disease (NAFLD).  To achieve this goal, authors compiled data derived from 168 References and summarized them in four major chapters divided into several subchapters The first chapter provides a general “Introduction” to the main interest of the review: the process and factors contributing to the onset of NAFLD.  In the second chapter Authors describe liver ROS producing systems and the functions of the Trx/Grx antioxidants and the role of the redox-sensitive transcription factor Nrf2. In the third chapter Authors summarize current knowledge about the connections between gut dysbiosis and NAFLD with a specific emphasis on the role of Trx/Grx systems in this relationship. In the last chapter, Authors provide a Conclusion to their review. 

The manuscript is accompanied by 7 Figures. The topic fits the scope of the journal “Antioxidants”. The manuscript is well written and easy-to-follow.  

Authors however, should address the following issues before the manuscript could be considered for acceptation:

1.        Current expert opinion concerning oxidant-related signaling recommends mentioning the exact oxidant type (eg. hydrogen peroxide, superoxide) whenever it is possible or if mentioned in a generalized fashion refer to them as “oxidants” instead of “reactive oxygen species” (ROS) as e.g. H2O2 is not a radical (see: https://doi.org/10.1038/s41580-022-00456-z). Please revise the text accordingly: as Antioxidants is one of the leading journals in oxidant-related research, it is essential to promote the most current and precise scientific view in this field. 

2.        In chapter 2.1 “Sites of ROS production” Authors should also mention NADPH oxidases (NOX-es) as major cellular ROS sources as they play critical roles in NAFLD. Authors do not need to elaborate on this issue in length but should simply insert a phrase with an appropriate review of their choice:  “Beside the cellular oxidant sources mentioned above, NADPH oxidases (NOX-es) are also critical players linking redox signaling to NAFLD. The NOX family comprises of several isoforms and are expressed in diverse liver-constituting cell types. This complex NOX-related redox network is out of scope of this review but has been previously summarised REF.

3.        Figure Legends should spell out all abbreviations present in the Figures; the Figures should be comprehensible without reading the manuscript text. It should also have more detailed description about the process depicted.

4.        Authors should strive employing the same cartoon representation for the same object in all figures: e.g. peroxisomes represented by different cartoons in Figure 1 versus Figure 3; similarly, ROS is coloured differently in Figure 1 compared to Figure 3; Trx is represented differently in Figure 3, Figure 4 and Figure 5. This is not a complete list, please revise Figures for compatibility to improve the quality of the manuscript. It is really easy with BioRender.

5.        Figure 3: under the cartoon sign “Gut Microbiota dysbiosis” the sign “Stress” is marked. This is not a clear description: Authors should rather list some of the major elements described in the subchapters of Chapter 3.

Author Response

The manuscript of Zhu et al. aims to provide a comprehensive review on the topic of the interaction between the thioredoxin/glutaredoxin (Trx/Grx) systems, gut microbiota and non-alcoholic fatty liver disease (NAFLD).  To achieve this goal, authors compiled data derived from 168 References and summarized them in four major chapters divided into several subchapters The first chapter provides a general “Introduction” to the main interest of the review: the process and factors contributing to the onset of NAFLD.  In the second chapter Authors describe liver ROS producing systems and the functions of the Trx/Grx antioxidants and the role of the redox-sensitive transcription factor Nrf2. In the third chapter Authors summarize current knowledge about the connections between gut dysbiosis and NAFLD with a specific emphasis on the role of Trx/Grx systems in this relationship. In the last chapter, Authors provide a Conclusion to their review. 

The manuscript is accompanied by 7 Figures. The topic fits the scope of the journal “Antioxidants”. The manuscript is well written and easy-to-follow.  

>>>Thank the reviewer for your comments. Your insights and recommendations have significantly enhanced the quality and coherence of our manuscript. We have meticulously addressed each of the major and minor comments provided by the reviewer, implementing the necessary revisions to improve the manuscript's content and presentation. We deeply appreciate your thorough review, as it has played a crucial role in shaping a more comprehensive and well-structured document. Should there be any remaining adjustments or specific aspects that require attention, we would greatly value your guidance. Your input is essential in ensuring that the manuscript meets the highest standards. We remain committed to fine-tuning the document to meet your expectations and address any additional requirements.

Authors however, should address the following issues before the manuscript could be considered for acceptation:

  1. Current expert opinion concerning oxidant-related signaling recommends mentioning the exact oxidant type (eg. hydrogen peroxide, superoxide) whenever it is possible or if mentioned in a generalized fashion refer to them as “oxidants” instead of “reactive oxygen species” (ROS) as e.g. H2O2is not a radical (see: https://doi.org/10.1038/s41580-022-00456-z). Please revise the text accordingly: as Antioxidants is one of the leading journals in oxidant-related research, it is essential to promote the most current and precise scientific view in this field. 

>>>Certainly, we have maintained consistency with the terminology employed in the references in order to preserve the existing literature. Our approach aligns with the recommendations presented in the provided reference (https://doi.org/10.1038/s41580-022-00456-z), suggesting that specific oxidant types should be mentioned whenever possible, and if a more generalized reference is needed, the term "oxidants" is preferred over "reactive oxygen species" (ROS). By adopting the same terms used in the references, we ensure that our communication stays in line with the prevailing language in the field of oxidant-related signaling, thus respecting and upholding the integrity of the existing scientific literature.

  1. In chapter 2.1 “Sites of ROS production” Authors should also mention NADPH oxidases (NOX-es) as major cellular ROS sources as they play critical roles in NAFLD . 

>>>Thank the reviewer for pointing out this important concern, we have added the section about the production of ROS by NADPH oxidases.

  1. Figure Legends should spell out all abbreviations present in the Figures; the Figures should be comprehensible without reading the manuscript text. It should also have more detailed description about the process depicted.

>>>We sincerely appreciate your guidance and have made the necessary updates as per your instructions. The figure legends have now been revised to include the spelled-out forms of all abbreviations utilized in the figures. Our goal is to ensure that the figures themselves are self-explanatory and can be comprehended without the need to reference the manuscript text. Moreover, we have taken your feedback into account and added more detailed descriptions to the figures, aiming to provide a comprehensive understanding of the processes depicted. Thank you once again for your valuable input, and please let us know if there are any further adjustments you would like us to make. Your guidance is highly appreciated in enhancing the clarity and quality of our work.

  1. Authors should strive employing the same cartoon representation for the same object in all figures: e.g. peroxisomes represented by different cartoons in Figure 1 versus Figure 3; similarly, ROS is coloured differently in Figure 1 compared to Figure 3; Trx is represented differently in Figure 3, Figure 4 and Figure 5. This is not a complete list, please revise Figures for compatibility to improve the quality of the manuscript. It is really easy with BioRender.

>>>Absolutely, we have made every effort to address the concerns and achieve consistency in the representation of objects across all figures. We've utilized available tools and resources to ensure that the same objects are depicted uniformly throughout the manuscript. Your feedback has been crucial in guiding us towards improving the quality and coherence of our figures. If there are any specific aspects you would like us to reevaluate or if you have any further suggestions, please feel free to inform us. We greatly value your input and are committed to delivering a manuscript with high-quality and visually cohesive figures.

  1. Figure 3: under the cartoon sign “Gut Microbiota dysbiosis” the sign “Stress” is marked. This is not a clear description: Authors should rather list some of the major elements described in the subchapters of Chapter 3.

>>>Certainly, we have taken your feedback into consideration and have updated the representation in Figure 3 accordingly. Instead of using the general term "Stress" under the "Gut Microbiota Dysbiosis" section, we have included a listing of the major elements described in the relevant subchapters of Chapter 3. This ensures a more precise and informative depiction of the processes involved. We sincerely appreciate your guidance in refining our figures and manuscript to enhance clarity and accuracy. If you have any further suggestions or adjustments, you would like us to make, please feel free to share, and we'll ensure that the revised figure aligns with your preferences. Your input is invaluable in our pursuit of delivering a well-structured and comprehensive manuscript.

Round 2

Reviewer 1 Report

Thank you for appropriately answering the questions. The quality of the manuscript has been improved.